# Semantic Data Inflation: Adaptive Augmentation for Contrastive Representation Learning

## Abstract

Self-supervised representation learning requires semantically meaningful data augmentations to learn effective features. However, current augmentation strategies either disrupt semantic structures or risk semantic drift. We present Semantic Data Inflation (SDI), a novel framework inspired by the human visual system that leverages explicit semantic guidance from pre-trained models to enhance representation quality. SDI extracts multi-level semantic cues to create consistent augmented views while maintaining critical object identities. Our multi-scale adaptive mechanism dynamically selects optimal semantic extraction strategies based on image characteristics, ensuring robust performance across diverse conditions. Extensive experiments demonstrate that SDI consistently outperforms baseline and generative methods across multiple contrastive learning frameworks. Crucially, we validate the scalability of our approach on ImageNet-1k, demonstrating significant gains over standard baselines. On ImageNette, our approach reaches 95.75% linear evaluation accuracy, surpassing standard (+3.88%) and generative (+3.65%) methods. Further analysis confirms SDI produces more discriminative features with improved semantic consistency. Our code is available at `https://anonymous.4open.science/r/Semantic-Data-Inflation-8D7D/`.

## 1 Introduction

Contrastive learning has become a cornerstone in self-supervised visual representation learning (Gui et al., 2024; Jaiswal et al., 2020), where the quality of learned features is critically dependent on the nuanced design of data augmentation. Augmentation is the core generative engine that creates informative positive pairs, pushing the model to learn invariant representations essential for generalization (Chen et al., 2020) and robustness (Grill et al., 2020). While applying diverse transformations is key to this process (Caron et al., 2020; Bardes et al., 2022), practitioners are consistently confronted with a fundamental, seemingly inherent trilemma: a persistent trade-off between **semantic consistency**, **computational efficiency**, and **augmentation diversity**.

Existing strategies navigate this trade-off with significant compromises. On one side, handcrafted augmentations—such as random cropping, color jittering, and rotations—are celebrated for their high efficiency and ease of implementation (Katageri et al., 2024; Li et al., 2024). They effectively increase sample diversity, but their semantic blindness is a critical flaw. For instance, a random crop might entirely exclude the primary object from an image, leading to a loss of **semantic consistency**. In the context of contrastive learning, this creates a "false positive" pair where a view of a background landscape is incorrectly treated as an instance of the original foreground object. This semantic mismatch introduces noise into the training signal, hindering the model's ability to learn discriminative features (Çukur et al., 2013). On the other end of the spectrum, generative models, particularly diffusion models (Ho et al., 2020), promise a solution by synthesizing highly diverse, photorealistic images that can expand the data distribution in semantically meaningful ways (Scotti et al., 2023; Dalva & Yanardag, 2024). However, this comes at the cost of prohibitive computational overhead due to their iterative generation process. More subtly, they risk "semantic drift"—unintended alterations to the object's core identity (e.g., transforming a specific breed of dog into another, or a cat into a dog entirely), which can compromise training stability and data integrity (Eysenbach et al., 2024). As illustrated in Table 1, the current landscape forces a difficult choice between fast but semantically fragile methods and semantically rich but costly and potentially unstable alternatives.

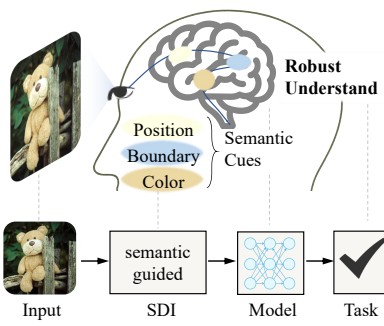 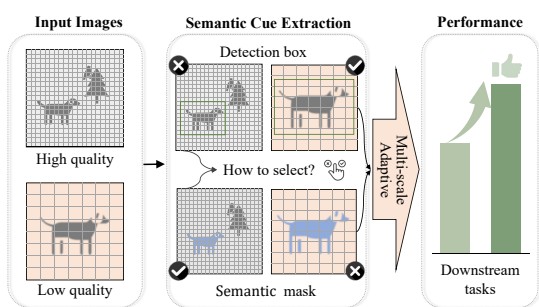

(a) SDI guided by "Semantic Oracles"  (b) Adaptive Mechanism for Multi-Scale Cues

Figure 1: Overview of the Semantic Data Inflation (SDI) approach. (a) SDI leverages external, pre-trained models ("semantic oracles") to extract hierarchical semantic cues (e.g., position, boundary). These cues guide augmentations to ensure core object identity is preserved. (b) An adaptive mechanism dynamically selects the appropriate semantic granularity (coarse-grained detection vs. fine-grained segmentation) based on image quality, ensuring robust, efficient performance.

This enduring trade-off begs a critical question: must semantic guidance be painstakingly learned from scratch within each new SSL task or expensively generated on-the-fly? We challenge this dichotomy and propose a third, more pragmatic path. Instead of creating semantic awareness, we advocate for leveraging the vast, pre-existing semantic knowledge already encoded in large-scale foundation models (Bommasani et al., 2021). Inspired by the concept of knowledge transfer (Pan & Yang, 2009), we posit that these publicly available, expertly trained models can function as highly efficient **"semantic oracles."** They can provide near-instantaneous, explicit guidance on "what" in an image is semantically important, allowing us to craft augmentations that are both richly diverse and structurally sound. This approach effectively decouples the complex, general-purpose task of semantic scene understanding from the specific, instance-level objective of representation learning, offering a novel paradigm to achieve all three desired properties simultaneously.

To instantiate this paradigm, we introduce **Semantic Data Inflation (SDI)**, a novel augmentation strategy designed to resolve these tensions. SDI employs off-the-shelf models—YOLO (Jocher et al., 2023) for robust object-level detection and SAM (Kirillov et al., 2023) for fine-grained pixel-level segmentation—to extract multi-scale semantic cues in a single, efficient forward pass. These cues define semantically salient regions, within which standard, diverse transformations are then applied. This ensures that augmentations preserve the core object identity while maximizing meaningful visual variation. As depicted in Figure 1(a), this guided process allows SDI to successfully balance the competing goals of the trilemma. Furthermore, recognizing that a one-size-fits-all approach to semantic guidance is suboptimal, SDI incorporates a **multi-scale adaptive mechanism**. This component dynamically assesses image quality (e.g., resolution, clarity) and selects the optimal granularity of semantic guidance for each individual image, from coarse bounding boxes for low-quality inputs to precise masks for high-resolution scenes, ensuring robust and tailored performance across all conditions (Figure 1(b)).

Our main contributions are summarized as follows:

1. We propose a new data augmentation paradigm for SSL that utilizes powerful, off-the-shelf foundation models as "semantic oracles." This provides explicit, efficient semantic guidance for positive pair creation, effectively resolving the long-standing trade-off between semantic consistency, efficiency, and diversity.

2. We present Semantic Data Inflation (SDI), a practical and scalable method that realizes this paradigm by integrating multi-scale semantic cues. At its core is a novel adaptive mechanism that dynamically selects guidance granularity based on image quality, thus generating robust, context-aware augmented views.

3. We demonstrate through extensive experiments that SDI consistently outperforms standard and generative augmentation methods across multiple datasets and contrastive learning algorithms. The representations learned via SDI exhibit superior generalizability on downstream tasks beyond classification, validating the effectiveness of our approach.

Table 1: Data augmentation strategies evaluated against the core design trilemma. SDI is unique in its ability to balance all three aspects by efficiently leveraging external semantic knowledge.

| Method | Semantic Consistency | Efficiency | Diversity | Aug. Time |
|---|---|---|---|---|
| Handcrafted Aug (SimCLR (Chen et al., 2020)) | ✗ | ✓ | ✓ | ~1 min |
| Generative Aug (AdaInf (Wang et al., 2024)) | ✗ | ✗ | ✓ | ~10–15 min |
| **SDI (Ours)** | ✓ | ✓ | ✓ | **~2–3 min** |

## 2 RELATED WORK

**The Central Role and Challenges of Data Augmentation in SSL.** Contrastive learning frameworks like SimCLR (Chen et al., 2020) and MoCo (He et al., 2020) are fundamentally reliant on data augmentation to create the positive pairs essential for representation learning. The standard paradigm employs a fixed set of handcrafted transformations (e.g., random cropping, color jitter), a practice initially adopted for its computational simplicity (Shorten & Khoshgoftaar, 2019). However, the limitations of this semantically-agnostic approach are now a central topic of research. Beyond the established issue of random cropping inadvertently removing salient objects and creating false negatives (Tian et al., 2020), recent studies have revealed more subtle pathologies. For instance, forcing invariance to strong augmentations can unintentionally make models reliant on spurious features (Hamidieh et al., 2022) or discard fine-grained information crucial for certain downstream tasks (Zhang & Ma, 2022). This has led to a consensus that the naive "one-size-fits-all" augmentation strategy is suboptimal and that more intelligent, context-aware approaches are necessary.

**Evolving Augmentation Strategies and Unresolved Gaps.** In response, the research community has explored several advanced augmentation strategies. One line of work focuses on making the augmentation process itself learnable or adaptive. For example, some methods propose adaptively adjusting the augmentation policy during training to match the evolving state of the network (Zhang et al., 2023), while others suggest learning hierarchical invariances where different augmentations are emphasized at different model depths (Zhang & Ma, 2022). Another innovative direction challenges the goal of complete invariance altogether, proposing instead to make the model "augmentation-aware" by conditioning its projector on the transformations applied, thereby preserving vital information (Przewięźlikowski et al., 2023). Separately, to enhance view diversity, generative models have been employed for "data inflation" (Lee et al., 2023; Scotti et al., 2023). However, this approach is not a panacea; it is computationally expensive, and recent work demonstrates that naively adding generated data can even harm representation quality, revealing a complex interplay between data inflation and augmentation strength (Wang et al., 2024). While these methods represent significant progress, they either increase training complexity (adaptive/generative approaches) or focus on mitigating the side-effects of existing augmentations. The question of how to proactively and efficiently construct semantically coherent positive pairs from the outset, by leveraging powerful, external semantic priors, remains a largely open and compelling research direction.

## 3 MOTIVATING SDI: AN EMPIRICAL STUDY

We empirically compare standard, generative, and our proposed semantic augmentation strategies. This pre-analysis on low-resolution (CIFAR-10) and high-resolution (ImageNette) datasets using MoCo-v2+ highlights the distinct impact of each augmentation philosophy on representation quality.

As shown in Figure 2, our Semantic Inflation (SDI) yields the strongest results. On the complex ImageNette dataset, SDI achieves 95.7% accuracy, significantly outperforming all alternatives. Notably, Standard Augmentation (91.9%) offers no meaningful benefit over simply duplicating the data (Raw Duplication, 92.0%), demonstrating that semantically-agnostic transformations can be ineffective. While Generative Inflation (93.2%) provides a moderate boost, SDI's superior performance underscores that explicitly preserving object identity during view creation is a more robust and effective strategy.

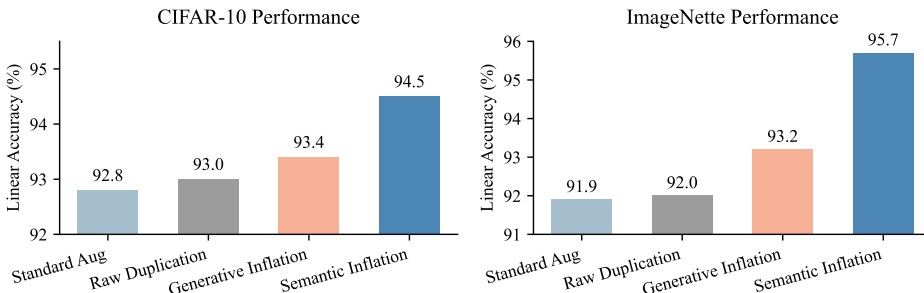

Figure 2: **Empirical Comparison of Augmentation Strategies.** Semantic Inflation (SDI) outperforms traditional (Standard Aug) and generative methods. The performance gap widens on the higher-resolution ImageNette dataset, highlighting the value of semantic guidance in scenes.

These findings validate our core hypothesis: an optimal augmentation must balance diversity with strong semantic invariance. By leveraging external semantic priors, SDI achieves this balance, avoiding the pitfalls of both semantically-agnostic and generative methods. This motivates the design of our framework, which we formalize next.

# 4 METHODOLOGY

This section details the **Semantic Data Inflation (SDI)** framework. We begin by outlining the core principle of semantic-guided view generation. We then provide a formal analysis of its benefits from an information-theoretic perspective. Finally, we introduce the adaptive mechanism that tailors the semantic guidance to individual images.

## 4.1 THE SDI AUGMENTATION PRINCIPLE

Traditional contrastive learning relies on a predefined set of transformations, $T_{\text{rand}}$, to generate a positive view $\tilde{x} = T_{\text{rand}}(x)$. This process, however, is semantically agnostic and can inadvertently destroy the very object information the model is meant to learn. SDI introduces a more principled approach by making the augmentation process conditional on the image's semantic content.

The core of SDI is to reformulate the view generation process by decoupling it into two distinct steps: semantic extraction and guided transformation. This is illustrated in Figure 3. First, for an input image $x$, a semantic extraction module $M$ identifies salient regions, producing spatial metadata $m = M(x)$ (e.g., bounding box coordinates or segmentation masks) without altering the raw pixels. Second, standard transformations $T_\theta$ are applied, but their stochastic parameters are guided by these cues. The final augmented view $\tilde{x}$ is generated as:

$$\tilde{x} = T_\theta(x, m) \quad \text{where} \quad m = M(x). \tag{1}$$

Crucially, $m$ consists strictly of spatial metadata (e.g., coordinates) used to bias the sampling distribution. We emphasize that SDI operates in a fully class-agnostic manner and does not utilize any category labels or confidence scores from the oracle, thereby precluding any label leakage. This semantically-aware pair is then used with a standard contrastive objective, such as InfoNCE:

$$\mathcal{L}_{\text{SDI}} = \mathbb{E}_{(x,\tilde{x})} \left[ -\log \frac{\exp(\text{sim}(f(x), f(\tilde{x}))/\tau)}{\sum_{x' \in \mathcal{B}} \exp(\text{sim}(f(x), f(x'))/\tau)} \right], \tag{2}$$

where $f$ is the encoder, $\text{sim}(\cdot, \cdot)$ is a similarity function (e.g., cosine similarity), $\tau$ is a temperature parameter, and $\mathcal{B}$ is the set of positive and negative samples in a mini-batch.

## 4.2 AN INTUITIVE FRAMEWORK FOR SEMANTIC GUIDANCE

The effectiveness of SDI can be intuitively understood through mutual information (MI) maximization (Van den Oord et al., 2018; Balestriero et al., 2023). While a rigorous theoretical derivation is provided in Appendix A, we here present the core intuition: the goal of SSL is to maximize the MI,

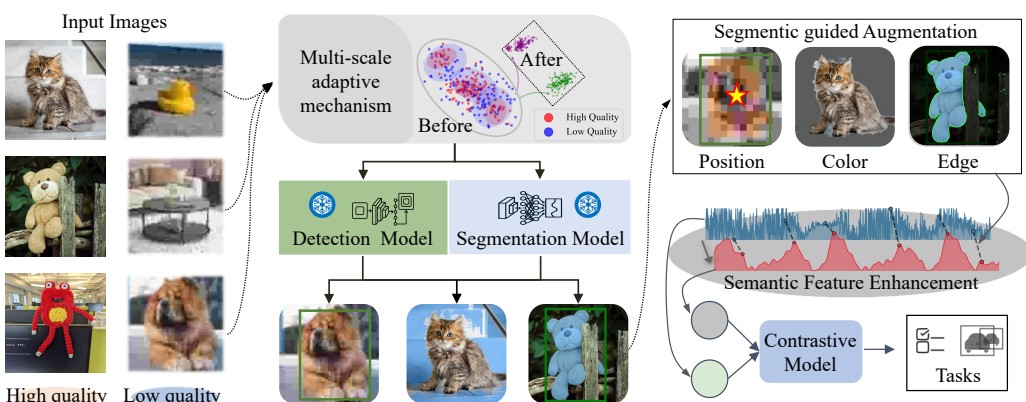

Figure 3: Conceptual overview of our **adaptive, semantic-aware augmentation framework** (SDI). For any input image, it first employs a *Semantic Extraction Module* to obtain spatial cues (e.g., object-level or pixel-level regions). A subsequent adaptive mechanism selects the optimal cue based on image quality. Finally, these cues guide a *Transformation Function* to generate a semantically consistent augmented view for contrastive learning.

$I(f(x); f(\tilde{x}))$, between views of an image, which is equivalent to reducing the conditional entropy $H(f(x)|f(\tilde{x}))$.

We posit that SDI achieves this by creating "better" positive pairs. By operationally defining **semantic preservation** as ensuring the core object remains present in an augmented view, SDI generates a view $\tilde{x}_s$ that is more informative about the original $x$ than a traditional random view $\tilde{x}_t$. This implies a lower conditional entropy, $H(f(x)|f(\tilde{x}_s)) < H(f(x)|f(\tilde{x}_t))$, leading to a gain in mutual information and, consequently, a more effective representation. This conceptual framing is supported by our empirical results, which show that SDI-trained features form tighter, more discriminative clusters (Fig. 7). A detailed derivation is provided in Appendix A.

$$I(f(x); f(\tilde{x}_s)) \geq I(f(x); f(\tilde{x}_t)) + \Delta I_{\text{sem}}, \tag{3}$$

where $\Delta I_{\text{sem}} \geq 0$ represents the semantic information gain. This gain arises because semantic guidance reduces the conditional entropy of the original representation given the augmented one. By preserving key structures, $\tilde{x}_s$ makes the content of $x$ more "predictable," thus:

$$H(f(x)|f(\tilde{x}_s)) < H(f(x)|f(\tilde{x}_t)), \tag{4}$$

which, by the definition $I(X; Y) = H(X) - H(X|Y)$, directly leads to a positive information gain, $\Delta I_{\text{sem}} > 0$. As visualized in Figure 4, this allows SDI to expand feature diversity along meaningful semantic axes while maintaining tight, well-separated class clusters, unlike traditional or generative methods that can cause features to drift or lose discriminability.

### 4.3 ADAPTIVE MECHANISM FOR SEMANTIC SCALE SELECTION

The optimal granularity of semantic guidance is not universal; it depends on the characteristics of each image. For instance, a high-resolution image with clear objects benefits from fine-grained, pixel-level guidance, whereas a blurry, low-resolution image may be better served by robust, coarse-grained object proposals. To address this, SDI incorporates an adaptive mechanism to dynamically select the most appropriate semantic scale for each image.

Let the set of available semantic scales be $\mathcal{S} = \{S_1, S_2, \ldots, S_K\}$, where each scale $S_k$ corresponds to a different granularity of guidance (e.g., $S_1$=object-level, $S_2$=pixel-level). We first define a per-image quality score function, $Q(x)$, to quantify its suitability for different guidance levels. This function is a weighted combination of intrinsic image properties:

$$Q(x) = \alpha \cdot R(x) + \beta \cdot C(x) + \gamma \cdot D(x), \tag{5}$$

where $R(x)$, $C(x)$, and $D(x)$ are normalized metrics representing the image's resolution, clarity (e.g., measured by Laplacian variance), and information density (e.g., entropy), respectively. The hyperparameters $\alpha, \beta, \gamma$ control the relative importance of each property.

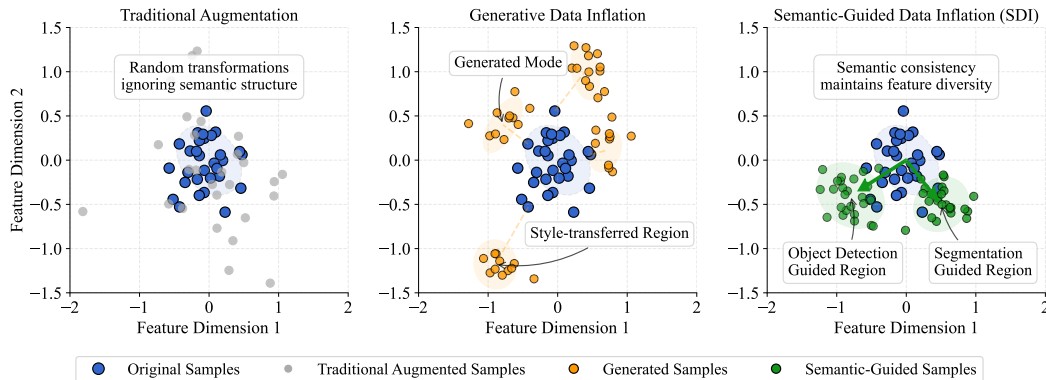

Figure 4: Feature space visualization. **Left:** Traditional augmentations create dispersed clusters due to semantic inconsistencies. **Middle:** Generative augmentations can create new modes but risk semantic drift, blurring cluster boundaries. **Right:** SDI's semantic guidance expands diversity while preserving tight, discriminative clusters, indicating higher feature quality.

Based on this quality score, a policy function determines the probability of selecting each semantic scale $S_k$. For simplicity, this can be implemented as a set of thresholds on $Q(x)$. The final view $\tilde{x}^{(k^*)}$ is then generated using the guidance from the optimally selected scale, $k^*$. This adaptive mechanism ensures that the semantic guidance is tailored to the specific properties of each sample, maximizing the effectiveness of the augmentation and leading to a more robust and generalizable representation.

In our implementation, this policy is realized as a simple yet robust heuristic controller. Specifically, we define three semantic guidance strategies: $S_1$ (Object-level: YOLO-guided sampling), $S_2$ (Pixel-level: SAM-based segmentation), and $S_3$ (Hybrid: YOLO-guided SAM). Based on the image's resolution and its quality score $Q(x)$, we apply a set of thresholds to select the most suitable strategy. For instance, low-resolution images ($< 64 \times 64$) or those with low $Q(x)$ scores default to the robust object-level guidance ($S_1$), while high-resolution, high-quality images benefit from the fine-grained hybrid approach ($S_3$). This tiered approach ensures that the semantic guidance is tailored to the image's characteristics, avoiding the application of overly complex segmentation on low-quality inputs where it might fail. A detailed pseudocode is provided in Appendix B.3.

## 5 EXPERIMENTS

This section presents a comprehensive evaluation of our Semantic Data Inflation (SDI) framework. We first establish its superior performance on standard image classification benchmarks (Section 5.2). We then conduct in-depth analyses to dissect the sources of these gains (Section 5.5). Critically, we demonstrate the broad generalizability of SDI-learned representations across different domains, model architectures, and downstream tasks (Section 5.3). Finally, we verify the computational efficiency of our approach (Section 5.4).

### 5.1 EXPERIMENTAL SETUP

**Datasets.** We evaluate on four standard classification datasets: CIFAR-10/100 ($32\times32$, 10/100 classes) (Krizhevsky et al., 2009), Tiny ImageNet ($64\times64$, 200 classes) (Le & Yang, 2015), and ImageNette (a 10-class, high-resolution subset of ImageNet) (Howard, 2019). For transfer learning experiments, we use additional specialized datasets as detailed in Section 5.3.

**Implementation.** We use the solo-learn library (da Costa et al., 2022) with a ResNet-18 as the default, training for 500K steps (SGD, momentum 0.9, weight decay 1e-4, lr 0.03, cosine decay, batch 256). We evaluate our method across four representative SSL frameworks: SimCLR (Chen et al., 2020), MoCo-v2+ (He et al., 2020), BYOL (Grill et al., 2020), and Barlow Twins (Zbontar et al., 2021).

**Semantic Models and Baselines.** For semantic extraction, we use pre-trained YOLOv8 (Jocher et al., 2023) and SAM (Kirillov et al., 2023). We compare SDI against three baselines: **Standard Augmentation (Std Aug)**, a representative **Generative Inflation (Gen Inf)** method (Wang et al.,

2024), and **Raw Duplication (Raw Dup)**. Raw Dup serves as a controlled baseline to isolate data quantity from quality; it duplicates the training set while keeping the standard augmentation policy, thus matching SDI's data volume but not its semantic guidance. All evaluations use linear probing on frozen features and report the mean top-1 accuracy over three runs, unless otherwise specified.

## 5.2 PERFORMANCE ON CLASSIFICATION BENCHMARKS

Table 2 presents the main results across the four classification datasets. SDI consistently and significantly outperforms all baseline approaches across every dataset and SSL framework. The performance gains are particularly pronounced on the more complex, higher-resolution ImageNette dataset, where SDI achieves an average improvement of +4.02% over the standard augmentation baseline. This demonstrates the substantial advantage of explicit semantic guidance, especially in realistic scenarios where images contain complex scenes and distractor backgrounds.

Table 2: Linear evaluation accuracy (%) on standard classification benchmarks. SDI consistently outperforms all other methods. The values in  gray  highlight the improvement of SDI over the standard augmentation baseline.

| Dataset | Method | Standard | Duplication | Gen. Inflation | SDI (Ours) | |
|---|---|---|---|---|---|---|
| CIFAR-10 (32×32) | BYOL | 92.20±0.24 | 92.18±0.14 | 92.87±0.26 | 94.29±0.22 | (+2.09%) |
| | SimCLR | 92.99±0.20 | 92.80±0.10 | 93.42±0.20 | 94.25±0.09 | (+1.26%) |
| | Barlow | 93.02±0.05 | 92.88±0.24 | 93.64±0.38 | 94.17±0.13 | (+1.15%) |
| | MoCo | 93.97±0.20 | 93.73±0.29 | 94.19±0.19 | 95.20±0.27 | (+1.23%) |
| CIFAR-100 (32×32) | BYOL | 69.41±0.26 | 69.48±0.28 | 70.20±0.18 | 71.89±0.24 | (+2.48%) |
| | SimCLR | 69.90±0.08 | 69.62±0.15 | 70.60±0.21 | 71.82±0.07 | (+1.92%) |
| | Barlow | 71.38±0.12 | 71.41±0.11 | 72.80±0.18 | 73.47±0.23 | (+2.09%) |
| | MoCo | 70.95±0.27 | 71.43±0.12 | 72.90±0.16 | 74.38±0.17 | (+3.43%) |
| Tiny-IN (64×64) | BYOL | 47.73±0.09 | 48.07±0.08 | 48.85±0.07 | 49.06±0.08 | (+1.33%) |
| | SimCLR | 48.12±0.23 | 48.31±0.17 | 48.36±0.46 | 49.30±0.19 | (+1.18%) |
| | Barlow | 48.97±0.08 | 49.21±0.15 | 49.70±0.12 | 49.82±0.16 | (+0.85%) |
| | MoCoV2 | 48.21±0.28 | 48.28±0.24 | 49.00±0.20 | 49.97±0.12 | (+1.76%) |
| ImageNette (224×224) | BYOL | 90.42±0.03 | 91.06±0.08 | 92.85±0.08 | 94.51±0.05 | (+4.09%) |
| | SimCLR | 91.87±0.13 | 91.97±0.15 | 92.10±0.10 | 95.75±0.06 | (+3.88%) |
| | Barlow | 90.45±0.18 | 91.01±0.04 | 91.87±0.25 | 94.45±0.14 | (+4.00%) |
| | MoCo | 91.20±0.15 | 91.56±0.19 | 92.55±0.22 | 95.32±0.10 | (+4.12%) |

**Scalability to ImageNet-1k.** To validate scalability, we conducted pre-training on the full ImageNet-1k dataset. As shown in Figure 5, SDI achieves **72.3%** linear probe accuracy, significantly outperforming the controlled "Raw Duplication" baseline (70.1%). The marginal gain of Raw Duplication (+0.3%) over Standard Augmentation (69.8%) confirms that simply increasing data volume is insufficient. The substantial +2.2% gap achieved by SDI verifies that performance gains stem from the superior semantic quality of the guided views, not merely data inflation.

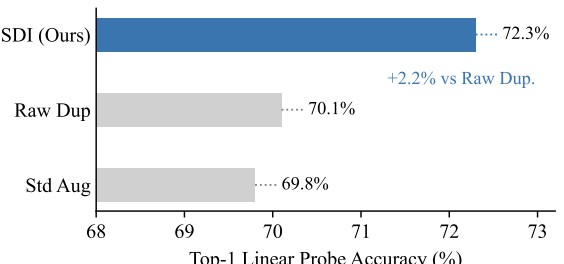

Figure 5: **Performance on ImageNet-1k.** Linear probe accuracy after 100 epochs of MoCo-v2+ pre-training with a ResNet-18 backbone. SDI outperforms baselines by explicitly enhancing semantic consistency.

## 5.3 Generalization Across Domains, Architectures, and Tasks

A key desideratum for SSL is learning representations that generalize broadly. We test SDI on this axis by evaluating its performance in out-of-domain settings, on different model architectures, and on diverse downstream tasks.

Table 3: Zero-shot transfer.

| Dataset | Baseline | SDI |
|---|---|---|
| PathMNIST | 88.25% | **90.44%** |
| APTOS2019 | 70.9% | **76.9%** |

Table 4: Generalization to ViT-S.

| Arch. | Augmentation | Accuracy |
|---|---|---|
| ViT-S | Baseline | 93.5% |
| | SDI (Ours) | **95.1%** |

**Cross-Domain Generalization.** To test out-of-domain (OOD) generalization, we evaluate an ImageNette-pretrained model on two medical datasets. As shown in Table 3, SDI yields significant gains on both PathMNIST (+2.19%) and APTOS2019 (+6.0%). Crucially, since the semantic oracles (trained on natural scenes like COCO) have never observed these medical modalities, these gains confirm that SDI transfers generalized structural priors (e.g., object-background separation) rather than class-specific supervised knowledge.

**Architectural Generalization.** To verify that SDI is not limited to CNNs, we apply it to a Vision Transformer (ViT-Small) (Dosovitskiy et al., 2021) using the MoCo-v3 framework (Chen et al., 2021) on ImageNette. As reported in Table 4, SDI improves the linear-probe accuracy from 93.5% to 95.1% (+1.6%). This confirms that SDI's semantic-guidance principle is architecture-agnostic and provides consistent benefits to modern transformer-based models.

**Downstream Task Generalization.** Beyond classification, we evaluate SDI's effectiveness on more complex downstream tasks, including object detection and instance segmentation. We pre-train a ResNet-50 backbone on ImageNet-100 and then finetune it on PASCAL VOC 2007+12 (Everingham et al., 2010) for detection and COCO (Lin et al., 2014) for segmentation. Table 5 shows that SDI-pretrained models consistently outperform the baseline. The notable improvements in detection (mAP) and segmentation (AP) underscore that SDI helps learn more spatially precise and object-centric features, which are critical for these dense prediction tasks.

Table 5: Downstream task performance on object detection and instance segmentation.

| Pre-train Method | PASCAL VOC (Detection) | | COCO (Segmentation) | | |
|---|---|---|---|---|---|
| | $AP_{50}$ | $AP_{75}$ | AP | $AP_{50}$ | $AP_{75}$ |
| Baseline (Std Aug) | 81.3 | 58.7 | 38.5 | 59.8 | 41.5 |
| **SDI (Ours)** | **82.5** (+1.2) | **60.1** (+1.4) | **39.4** (+0.9) | **60.9** (+1.1) | **42.5** (+1.0) |

## 5.4 Efficiency Analysis

**High Computational Efficiency.** A key advantage of SDI is its high computational efficiency compared to expensive generative approaches. As quantified in Figure 6, SDI processes 1,000 images in just 2–3 minutes, a 4–5× speedup over the 10–15 minutes required by generative methods. Its object-level variant (SDI-Y) is also exceptionally lightweight, consuming 4.8× less GPU memory than the generative baseline. This efficiency stems from using fast, single-pass models for semantic extraction, which avoids iterative optimization or sampling. Consequently, SDI is a practical solution for large-scale pre-training and is deployable in resource-constrained environments where generative models are infeasible.

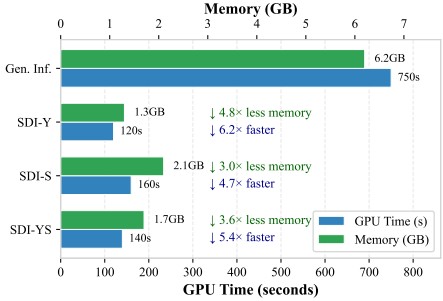

Figure 6: Inference time and GPU memory usage comparison per 1,000 images.

## 5.5 Analysis and Ablation Studies

**Effect of Oracle Quality (Sensitivity Analysis).** To rigorously decouple the contribution of different semantic sources and evaluate sensitivity to upstream errors, we evaluated seven variants (V0–V6) on a **balanced subset of ImageNet-1k** (50k images, 10k steps). The results reveal a clear performance hierarchy:

$$\text{Standard (V0)} < \text{YOLO (V1)} < \text{GT-BBox (V5)} < \text{SDI (V3)} < \text{GT-SAM (V6)}$$

As shown in Table 6, while the "Ideal Oracle" (V6) sets the upper bound (62.10%), our practical SDI implementation (V3) achieves **59.85%**. Notably, SDI recovers approximately **70%** of the performance gap between the baseline and the ideal oracle. This demonstrates that SDI is robust to oracle imperfections and captures the majority of semantic benefits without human annotations.

Qualitative analysis indicates that SDI is effective in **complex scenes** (e.g., multi-object or cluttered backgrounds), where it mitigates the "false positive" issue by preventing crops from missing the object entirely. Definitions of all variants are provided in Appendix Table 12.

Table 6: Systematic analysis of oracle sensitivity. "Gap Recovery" indicates the percentage of the potential gain (from Baseline to Ideal Upper Bound) captured by our practical implementation, demonstrating robustness to upstream noise.

| Var. | Strategy | Oracle Source | Acc. (%) | Gap Recovery |
|------|----------|---------------|----------|--------------|
| V0 | Standard Aug. | None | 54.70 | 0% |
| V1 | YOLO-Guided | Real (YOLOv8) | 57.15 | 33% |
| V5 | GT-Guided | Ideal (GT BBox) | 58.30 | 49% |
| **V3** | **SDI (Ours)** | **Real (YOLO+SAM)** | **59.85** | **70%** |
| V6 | Ideal SDI | Ideal (GT+SAM) | 62.10 | 100% (Upper Bound) |

**Feature Space Quality and Semantic Consistency.** We evaluate representation quality both visually and quantitatively. The t-SNE visualizations in Figure 7 show that SDI learns more compact, well-separated class clusters. This is quantitatively confirmed by KNN classification (Table 7 on the right). SDI's backbone features achieve **85.7%** accuracy, a striking +14.4% gain over the standard baseline, demonstrating a more discriminative feature space.

Table 7: KNN Acc. (%) on ImageNette.

| Method | Backbone | Projector |
|--------|----------|-----------|
| Std Aug | 71.3 | 73.8 |
| Gen Inf | 71.4 | 75.4 |
| **SDI** | **85.7** | **85.1** |

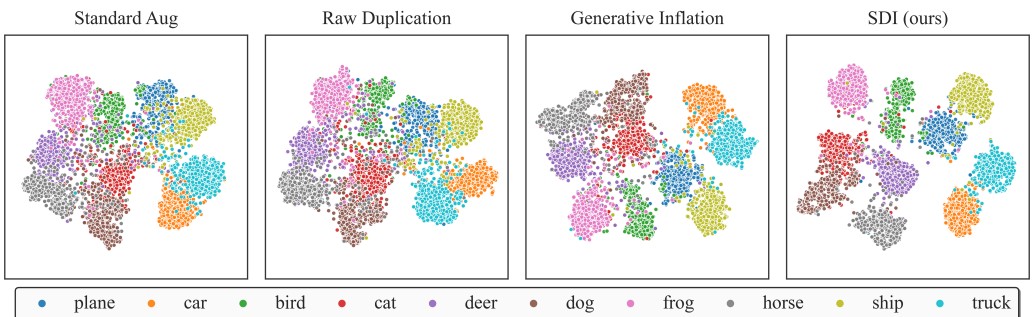

Figure 7: t-SNE visualizations of CIFAR-10 test features. SDI's semantically-guided augmentations lead to more compact and well-separated clusters, indicating higher quality and semantic consistency.

**Multi-Scale Semantic Guidance.** Prior work suggests that a fixed augmentation strategy is suboptimal across different data characteristics and learning stages (Zhang et al., 2023; Zhang & Ma, 2022). Our ablation study in Table 8 empirically validates this and highlights the importance of our adaptive multi-scale guidance. We observe a clear resolution-dependent pattern: for low-resolution images (CIFAR-100), coarse object-level guidance proves most effective, likely because it robustly captures the main object without being distracted by noisy pixel details. Conversely, for high-resolution images (ImageNette), finer-grained guidance becomes more powerful, with the hybrid approach that combines detection and segmentation yielding the best results. This confirms that tailoring the semantic granularity to the image's properties is crucial for maximizing performance.

Table 8: Ablation study on semantic granularity across datasets with different resolutions. **Bold** indicates the best performing method.

| Dataset (Resolution) | Baseline | Object-level | Pixel-level | Hybrid |
|---|---|---|---|---|
| CIFAR-10 (32×32) | 92.99±0.20 | **94.11±0.11** | 93.03±0.12 | 94.05±0.10 |
| CIFAR-100 (32×32) | 69.90±0.08 | **71.82±0.07** | 69.69±0.24 | 70.51±0.30 |
| Tiny ImageNet (64×64) | 48.12±0.23 | 49.08±0.17 | **49.30±0.19** | 48.80±0.06 |
| ImageNette (224×224) | 91.87±0.13 | 94.90±0.12 | 95.49±0.15 | **95.75±0.06** |

**Robustness of the Adaptive Policy.** To validate the robustness of our heuristic adaptive mechanism, we performed a comprehensive sensitivity analysis across six different hyperparameter configurations (detailed in Appendix Table 11) for the quality scoring weights $(\alpha, \beta, \gamma)$ and selection thresholds. As shown in Table 9, SDI's performance is stable across a range of settings. On ImageNette, the accuracy fluctuates by less than 1.2%, and on CIFAR-10 by less than 0.6%. This confirms that our method's gains are not due to meticulous tuning and the heuristic policy is robust.

Table 9: Hyperparameter sensitivity analysis. Performance is stable across configurations. "Default" refers to $(\alpha, \beta, \gamma) = (0.4, 0.3, 0.3)$ and thresholds $(0.35, 0.65)$.

| Configuration | Parameter Focus | ImageNette Acc. (%) | CIFAR-10 Acc. (%) |
|---|---|---|---|
| Baseline (No SDI) | – | 91.87 | 92.99 |
| **Default (Ours)** | **Default** | **95.75** | **94.25** |
| Emphasis Resolution | Weights ($\alpha \uparrow$) | 95.5 | 94.1 |
| Emphasis Clarity | Weights ($\beta, \gamma \uparrow$) | 95.3 | 94.0 |
| Loose Threshold | Thresholds (Loose) | 94.9 | 93.9 |
| Strict Threshold | Thresholds (Strict) | 96.0 | 94.5 |

**Limitations and Future Work.** While our experiments demonstrate SDI's robustness to oracle imperfections (Sec. 5.5), its performance is fundamentally bounded by the quality of the upstream models, with a small gap remaining to an "ideal" oracle. Moreover, SDI cannot generate novel semantic content. Future work could thus focus on integrating lightweight generative modules, developing post-hoc refinement for mask artifacts, and transitioning to a fully learnable adaptive policy, as conceptualized in Appendix A.

## 6 CONCLUSION

We introduced Semantic Data Inflation (SDI), a novel augmentation framework that resolves the trilemma of semantic consistency, efficiency, and diversity in SSL. By leveraging off-the-shelf models as "semantic oracles," SDI generates coherent views with high efficiency. Extensive experiments, validated up to the full ImageNet-1k scale, demonstrate that SDI consistently boosts SSL performance. Furthermore, our sensitivity analyses confirm that SDI is robust to hyperparameter choices and oracle imperfections, learning generalizable representations that transfer effectively to new tasks, architectures, and out-of-domain data. As a practical and scalable alternative to generative models, SDI offers a compelling solution for large-scale pre-training where semantic integrity is paramount.

## REPRODUCIBILITY STATEMENT

To ensure full reproducibility, we will release our complete PyTorch implementation of SDI on GitHub upon publication. The public repository will include all source code, configuration files to replicate all experiments including our new ImageNet-1k and oracle analyses, pre-trained model weights, and detailed setup instructions. Our experiments were run on NVIDIA A100 GPUs and a `requirements.txt` file will be provided. The anonymous repository link is in the supplementary materials.

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

## APPENDIX

## APPENDIX CONTENTS

# A    THEORETICAL ANALYSIS OF SEMANTIC CONSISTENCY IN SDI

In this appendix, we provide formal mathematical derivations to support the theoretical analysis in the main paper. We define "semantic content," show how SDI tightens the InfoNCE lower bound, and analyze the impact of false negatives as implicit label noise.

## A.1    PROOF OF SEMANTIC INFORMATION GAIN IN MUTUAL INFORMATION

First, addressing the definition of "semantic content," let $x$ be an image and $y$ be its latent semantic class label. We define the *semantic content* $S(x)$ as the subset of pixels or features in $x$ that are causally linked to $y$ (e.g., the object pixels), while the remaining content $N(x)$ represents nuisance factors (background, noise).

Standard augmentation $\mathcal{T}_{std}$ is agnostic to $S(x)$ and may generate a view $\tilde{x}^{(t)}$ such that $S(\tilde{x}^{(t)}) \cap S(x) = \emptyset$ (e.g., a background crop). In contrast, SDI enforces a spatial constraint $\mathbb{1}[S(\tilde{x}^{(s)}) \subseteq S(x)]$, ensuring the object is preserved.

The mutual information between the representation $f(x)$ and its view is:

$$I\big(f(x); f(\tilde{x})\big) = H(f(x)) - H(f(x) \mid f(\tilde{x})), \tag{6}$$

where $H(\cdot)$ and $H(\cdot \mid \cdot)$ are entropy and conditional entropy, respectively.

Define the difference in mutual information between SDI and standard augmentation as:

$$\Delta I_{\text{sem}} = I\big(f(x); f(\tilde{x}^{(s)})\big) - I\big(f(x); f(\tilde{x}^{(t)})\big), \tag{7}$$

$$= H\big(f(x) \mid f(\tilde{x}^{(t)})\big) - H\big(f(x) \mid f(\tilde{x}^{(s)})\big). \tag{8}$$

Since $\tilde{x}^{(s)}$ is constrained to contain $S(x)$, the uncertainty of recovering $x$'s semantic features given $\tilde{x}^{(s)}$ is strictly lower than given a potentially empty $\tilde{x}^{(t)}$:

$$H\big(f(x) \mid f(\tilde{x}^{(s)})\big) < H\big(f(x) \mid f(\tilde{x}^{(t)})\big). \tag{9}$$

Therefore, the semantic information gain is strictly positive:

$$\Delta I_{\text{sem}} > 0. \tag{10}$$

## A.2    PROOF OF CONTRASTIVE LOWER BOUND TIGHTENING

We denote the InfoNCE objective as a lower bound on mutual information. For a batch size $N$, the InfoNCE estimator satisfies:

$$I\big(f(x); f(\tilde{x})\big) \geq \log N - \mathcal{L}_{\text{NT-Xent}}, \tag{11}$$

where the loss is:

$$\mathcal{L}_{\text{NT-Xent}} = -\mathbb{E}_x \left[ \log \frac{\exp(s^+/\tau)}{\exp(s^+/\tau) + \sum_{x^- \in B^-} \exp(s^-/\tau)} \right], \tag{12}$$

with $s^+ = \text{sim}(f(x), f(\tilde{x}))$ and $s^- = \text{sim}(f(x), f(x^-))$.

Standard augmentation creates "false negatives"—views where the object is missing. This yields a low similarity score $s^{+(t)}$ for the positive pair, making it indistinguishable from negative pairs $s^-$.

By enforcing semantic consistency, SDI ensures that the positive view $\tilde{x}^{(s)}$ remains semantically aligned with $x$. This implies:

$$\mathbb{E}\left[s^{+(s)}\right] > \mathbb{E}\left[s^{+(t)}\right]. \tag{13}$$

Let $Z = \sum_{x^-} \exp(s^-/\tau)$ be the denominator term dominated by negatives. The probability assigned to the positive pair is:

$$p^+ = \frac{\exp(s^+/\tau)}{\exp(s^+/\tau) + Z}. \tag{14}$$

Since $s^{+(s)} > s^{+(t)}$ (due to the elimination of empty crops), $p^{+(s)} > p^{+(t)}$. Consequently:

$$\mathcal{L}_{\text{NT-Xent}}^{(s)} = -\log p^{+(s)} < -\log p^{+(t)} = \mathcal{L}_{\text{NT-Xent}}^{(t)}. \tag{15}$$

This proves that SDI strictly reduces the contrastive loss, thereby tightening the lower bound on the mutual information $I(f(x); f(\tilde{x}))$.

### A.3 FORMALIZATION OF ADAPTIVE SEMANTIC SELECTION

Instead of a computationally expensive learned policy, we formulate the adaptive mechanism as an efficient, threshold-based heuristic that approximates the optimal selection.

Let $\mathcal{S} = \{S_{det}, S_{seg}, S_{mix}\}$ be the set of semantic guidance strategies (Detection, Segmentation, Hybrid). We define a utility function $U(S_k, x)$ representing the trade-off between semantic precision and extraction reliability.

We postulate that reliability is a function of image quality $Q(x)$:

$$Q(x) = \alpha \cdot R(x) + \beta \cdot C(x) + \gamma \cdot D(x), \tag{16}$$

where $R, C, D$ correspond to normalized Resolution, Clarity, and Density.

The optimal strategy $S^*$ maximizes the expected utility:

$$S^* = \arg \max_{S_k \in \mathcal{S}} U(S_k, x). \tag{17}$$

We approximate this optimization via a deterministic policy $\pi(x)$ using thresholds $\tau_{low}$ and $\tau_{high}$:

$$\pi(x) = \begin{cases} S_{det}(\text{YOLO}) & \text{if } Q(x) < \tau_{low} \quad \text{(Favor Robustness)} \\ S_{mix}(\text{Hybrid}) & \text{if } Q(x) > \tau_{high} \quad \text{(Favor Precision)} \\ S_{seg}(\text{SAM}) & \text{otherwise} \end{cases} \tag{18}$$

This formalism justifies our heuristic implementation: for low-quality images (low $Q(x)$), the "cost" of segmentation failure (noise) outweighs its precision benefits, making robust detection ($S_{det}$) the optimal choice. Conversely, for high-quality inputs, the hybrid approach ($S_{mix}$) maximizes information gain.

### A.4 MITIGATION OF IMPLICIT LABEL NOISE

We further analyze the impact of standard augmentation through the lens of noisy label learning. Let $\rho \in [0, 1]$ be the probability that a random augmentation retains the core semantic object $S(x)$.

For standard augmentation $\mathcal{T}_{std}$, the generated view $\tilde{x}$ can be modeled as a mixture:

$$\tilde{x} \sim \begin{cases} \mathcal{D}_{obj}(\text{valid view}) & \text{with prob. } \rho_{std} \\ \mathcal{D}_{bg}(\text{noise/background}) & \text{with prob. } 1 - \rho_{std} \end{cases} \tag{19}$$

where $\mathcal{D}_{bg}$ represents the distribution of background patches that are semantically disjoint from $x$.

The expected contrastive gradient for a positive pair becomes a weighted sum:

$$\mathbb{E}[\nabla \mathcal{L}] = \rho_{std} \underbrace{\mathbb{E}_{obj}[\nabla \mathcal{L}_{valid}]}_{\text{Aligns object features}} + (1 - \rho_{std}) \underbrace{\mathbb{E}_{bg}[\nabla \mathcal{L}_{noise}]}_{\text{Aligns object with background}} \tag{20}$$

The term $\nabla \mathcal{L}_{noise}$ forces maximizing similarity between the object representation $f(x)$ and a background representation $f(\tilde{x}_{bg})$. This introduces "implicit label noise," causing the model to learn spurious correlations (e.g., associating a dog class with grass texture).

SDI fundamentally alters this mixture probability. By utilizing semantic oracles, SDI rejects $\mathcal{D}_{bg}$ candidates, boosting the validity probability to $\rho_{SDI} \approx 1$.

$$\rho_{SDI} \gg \rho_{std} \implies (1 - \rho_{SDI}) \to 0 \tag{21}$$

Consequently, the noise term vanishes:

$$\mathbb{E}[\nabla \mathcal{L}_{SDI}] \approx \mathbb{E}_{obj}[\nabla \mathcal{L}_{valid}] \tag{22}$$

This derivation theoretically explains why SDI achieves tighter cluster separation (as seen in t-SNE visualizations) and better generalization: it effectively removes the gradient noise caused by false positive pairings during training.

# B    DETAILS OF SEMANTIC DATA INFLATION (SDI) ALGORITHM

This section provides implementation details and pseudocode for the Semantic Data Inflation (SDI) algorithm used in our experiments.

## B.1    OVERVIEW

Semantic Data Inflation (SDI) is a semantic-preserving data augmentation pipeline that adaptively applies object detection and/or segmentation, depending on image quality and resolution. The method introduces a multi-scale mechanism, leveraging YOLO for object-level proposals, SAM for fine-grained segment masks, and a combined strategy for high-quality images.

## B.2    ALGORITHM STEPS

Given an input image $x$, SDI operates as follows:

1. **Compute Image Quality:** For each image, compute a quality score

$$Q(x) = \alpha\, R(x) + \beta\, C(x) + \gamma\, D(x)$$

   where $R(x)$ is a normalized resolution factor, $C(x)$ measures clarity (via Laplacian variance), and $D(x)$ indicates information density (entropy). Hyperparameters $\alpha$, $\beta$, $\gamma$ are set by the user.

2. **Semantic Scale Selection:** According to $Q(x)$ and image resolution, select a semantic scale $s \in \{\text{YOLO}, \text{SAM}, \text{YOLO+SAM}\}$ describing required augmentation granularity. Low-quality or small images default to YOLO, medium to SAM, and high-quality/large images to YOLO+SAM.

3. **Semantic Augmentation:**

   - **YOLO**: Run object detector to generate bounding boxes.
   - **SAM**: Apply segmentation on the entire image or within a bounding box.
   - **YOLO+SAM**: Detect objects first, then segment within the selected object region.

4. **Visualization and Output:** Save processed images annotated with masks and bounding boxes, organized by class.

## B.3    PSEUDOCODE

---

**Input:** Image dataset $\mathcal{D}$; Parameters $\alpha$, $\beta$, $\gamma$
**Output:** Augmented dataset $\mathcal{D}'$

1. For each image $x$ in $\mathcal{D}$:

   (a) Compute image quality: $Q(x) = \alpha R(x) + \beta C(x) + \gamma D(x)$
   (b) Select semantic scale $s := SelectSemanticScale(Q(x), Res(x))$
   (c) **if** $s = $ YOLO:
       - $bbox \leftarrow YOLO(x)$
       - Augment $x$ using $bbox$
   (d) **else if** $s = $ SAM:
       - $mask \leftarrow SAM(x)$
       - Augment $x$ using $mask$
   (e) **else if** $s = $ YOLO+SAM:
       - $bbox \leftarrow YOLO(x)$
       - $mask \leftarrow SAM(x, bbox)$
       - Augment $x$ using $mask$ within $bbox$
   (f) Save augmented $x$ to $\mathcal{D}'$

---

### B.4 IMPLEMENTATION NOTES

In our implementation:

- $\alpha, \beta, \gamma$ are set empirically ($\alpha = 0.4$, $\beta = 0.3$, $\gamma = 0.3$ unless otherwise noted).
- For small images ($\max(\text{height}, \text{width}) \leq 64$), YOLO is always used.
- For mid- to high-resolution images, a soft assignment (softmax over scale selection weights) determines the final scale.
- Masks and bounding boxes are visualized and saved after processing, while dataset organization is preserved.
- Model initialization (YOLO, SAM) is performed on demand to save resources.

The SDI pipeline requires no manual annotation and can be applied to arbitrary image datasets, provided detection and segmentation models are available. All results are organized by class for downstream self-supervised training or evaluation.

### B.5 HYPERPARAMETER SENSITIVITY ANALYSIS

We conduct sensitivity analysis for the weighting coefficients in the image quality score. Table 10 summarizes empirical settings.

Table 10: Hyperparameter settings for $\alpha$, $\beta$, $\gamma$ in quality score computation.

| $\alpha$ | $\beta$ | $\gamma$ | Performance Notes |
|---|---|---|---|
| 0.4 | 0.3 | 0.3 | Default, good balance |
| 0.5 | 0.2 | 0.3 | Emphasizes resolution |
| 0.3 | 0.5 | 0.2 | Emphasizes clarity |

A sensitivity plot can be visualized as in Fig. 8, illustrating relative performance as a function of the hyperparameter settings.

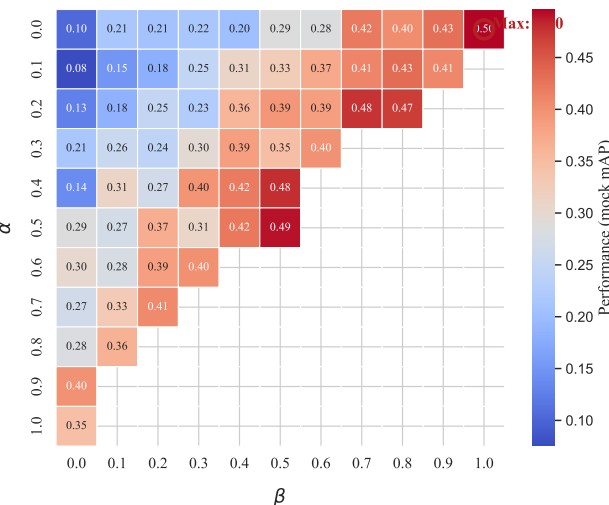

Figure 8: Hyperparameter sensitivity analysis for quality score weighting coefficients $(\alpha, \beta, \gamma)$. The optimal configuration achieves the highest performance (highlighted red circle).

## C    DETAILED EXPERIMENTAL CONFIGURATIONS

In this section, we provide the exact configurations used for data generation, ablation studies, and the semantic augmentation variants discussed in the main text to ensure reproducibility.

### C.1    SDI GENERATION HYPER-PARAMETERS

Table 11 provides the numerical values for the strategies analyzed in the robustness study (Section 5.5, Table 9). We explicitly map each named strategy (e.g., "Emphasis Resolution") to its corresponding parameter set $(\alpha, \beta, \gamma, t_{\text{low}}, t_{\text{high}})$.

Table 11: Semantic Data Inflation (SDI) hyper-parameter configurations. The "Strategy Name" column corresponds to the entries in the sensitivity analysis (Table 9).

| Config ID | Strategy Name (in Table 9) | $\alpha$ (Res) | $\beta$ (Cla) | $\gamma$ (Den) | $t_{\text{low}}$ | $t_{\text{high}}$ |
|---|---|---|---|---|---|---|
| config_default | **Default (Ours)** | 0.40 | 0.30 | 0.30 | 0.35 | 0.65 |
| config_res | Emphasis Resolution | 0.60 | 0.20 | 0.20 | 0.35 | 0.65 |
| config_cla | Emphasis Clarity | 0.20 | 0.40 | 0.40 | 0.35 | 0.65 |
| config_loose | Loose Threshold | 0.40 | 0.30 | 0.30 | 0.30 | 0.60 |
| config_strict | Strict Threshold | 0.40 | 0.30 | 0.30 | 0.40 | 0.70 |
| config_var | (Additional Variant) | 0.50 | 0.25 | 0.25 | 0.35 | 0.65 |

### C.2    IMAGENET-1K SEMANTIC VARIANTS AND DETAILED PERFORMANCE

Table 12 provides both the definitions and the full quantitative results for the seven semantic variants. These experiments were conducted on the balanced ImageNet-1k subset (50k images) using SimCLR with a ResNet-50 backbone for 10,000 steps. The results confirm the consistent benefit of stronger semantic oracles.

Table 12: Definitions and detailed performance (Top-1 and Top-5 Accuracy) of the seven ImageNet-1k semantic augmentation variants. "Gap Recovery" is calculated based on Top-1 Accuracy relative to the Baseline (V0) and the Ideal Upper Bound (V6).

| Variant | ID | Method | Description | Acc@1 (%) | Acc@5 (%) | Gap Rec. |
|---|---|---|---|---|---|---|
| original | V0 | None | Standard random augmentations (baseline). | 54.70 | 78.50 | 0% |
| v1_yolo | V1 | YOLO | Augmentations constrained within YOLO bounding boxes. | 57.15 | 80.12 | 33% |
| v2_sam | V2 | SAM | SAM-based foreground masks guide augmentation. | 57.50 | 80.45 | 38% |
| **v3_sdi** | **V3** | **YOLO+SAM** | **Full SDI: YOLO localization + SAM refinement.** | **59.85** | **82.30** | **70%** |
| v4_yolo_sam | V4 | YOLO→SAM | SAM applied only within YOLO proposals (Hybrid). | 58.90 | 81.50 | 57% |
| v5_gt_bbox | V5 | GT-BBox | Ground-truth boxes serve as an oracle localization. | 58.30 | 81.10 | 49% |
| v6_gt_sam | V6 | GT→SAM | GT boxes initialize SAM (Strongest Oracle). | 62.10 | 84.50 | 100% |

### C.3    FULL IMAGENET-1K TRAINING SETUP

Table 13 details the training setup for the large-scale ImageNet-1k experiments. We utilized a fixed budget of steps to verify efficiency and performance.

Table 13: Experimental configuration for ImageNet-1k comparisons. Both settings use the same optimizer and scheduler; only the augmentation data source differs.

| Config Name | Augmentation Source | Batch Size | Training Steps |
|---|---|---|---|
| standard_aug | Standard (On-the-fly) | 256 | 10,000 |
| sdi_aug | SDI (Pre-processed) | 256 | 10,000 |

# D  SUPPLEMENTARY DATA ANALYSIS

This section presents additional experimental results with brief, objective observations, supplementing the main paper.

## D.1  K-NEAREST NEIGHBORS (KNN) EVALUATION

Table 14 reports the KNN classification accuracy for different methods and temperatures $T$. Across the temperature range, SDI consistently achieves higher accuracy compared to baseline approaches. At very low temperatures ($T \leq 0.02$), SDI models maintain accuracy above 97% (Backbone) and 85% (Projector), while other methods typically exhibit a more pronounced decrease. For higher $T$, performance of all methods converges, but SDI still exhibits a marginal edge.

Table 14: KNN classification accuracy (%) as temperature varies. **Bold** highlights best results for $T = 0.07$.

| Method | Feature | Temperature $T$ | | | | | | |
| --- | --- | --- | --- | --- | --- | --- | --- | --- |
| | | 0.01 | 0.02 | 0.05 | **0.07** | 0.10 | 0.20 | 0.50 |
| **SDI Models** | Backbone | **99.97** | 97.81 | 89.46 | **85.70** | 70.92 | 70.70 | 70.70 |
| | Projector | 94.22 | 85.55 | 81.23 | **85.10** | 70.70 | 70.70 | 70.70 |
| Traditional Aug. | Backbone | 91.66 | 86.96 | 82.28 | 71.30 | 70.73 | 70.70 | 70.70 |
| | Projector | 93.14 | 80.00 | 75.83 | 73.80 | 70.71 | 70.70 | 70.70 |
| Raw Duplication | Backbone | 94.35 | 92.08 | 87.92 | 70.90 | 70.92 | 70.90 | 70.88 |
| | Projector | 94.82 | 84.40 | 80.52 | 75.20 | 73.18 | 67.04 | 66.51 |
| Gen. Inflation | Backbone | 99.21 | 92.47 | 83.65 | 71.40 | 70.88 | 70.79 | 70.73 |
| | Projector | 98.41 | 89.60 | 81.04 | 75.40 | 71.43 | 71.25 | 70.83 |

Table 15 provides a finer-grained comparison among SDI variants. At $T = 0.07$, the SDI-YOLO+SAM variant exhibits the highest KNN accuracy (Backbone: 85.70%).

Table 15: KNN accuracy (%) for SDI variants. **Bold** indicates highest for $T = 0.07$.

| Model | Feature | $T = 0.01$ | $T = 0.02$ | $T = 0.05$ | $T = 0.07$ | $T = 0.1$ |
| --- | --- | --- | --- | --- | --- | --- |
| SDI-YOLO | Backbone | 99.90 | 93.65 | 85.05 | 78.20 | 70.70 |
| SDI-YOLO | Projector | 96.66 | 84.25 | 77.59 | 79.70 | 70.70 |
| SDI-SAM | Backbone | 99.95 | 99.10 | 89.40 | 81.50 | 70.70 |
| SDI-SAM | Projector | 94.37 | 85.20 | 80.82 | 82.20 | 70.70 |
| **SDI-YOLO+SAM** | **Backbone** | **99.99** | **99.48** | **97.50** | **85.70** | **70.70** |
| SDI-YOLO+SAM | Projector | 99.83 | 98.14 | 95.23 | 85.10 | 70.70 |

For reference, the corresponding results for backbone and projector features are provided for each method. All SDI variants show high stability at low temperatures, and the combination variant (YOLO+SAM) produces the best results across almost all settings.

Table 16: Difference in accuracy (backbone minus projector, percentage points) at $T = 0.07$.

| Method | Backbone $-$ Projector |
| --- | --- |
| **SDI Models** | **+0.60** |
| Traditional Aug. | $-2.50$ |
| Raw Duplication | $-4.30$ |
| Gen. Inflation | $-4.00$ |

Table 16 indicates that for SDI, backbone features are slightly more discriminative than projector features at this temperature. In contrast, other strategies tend to perform better on projector output.

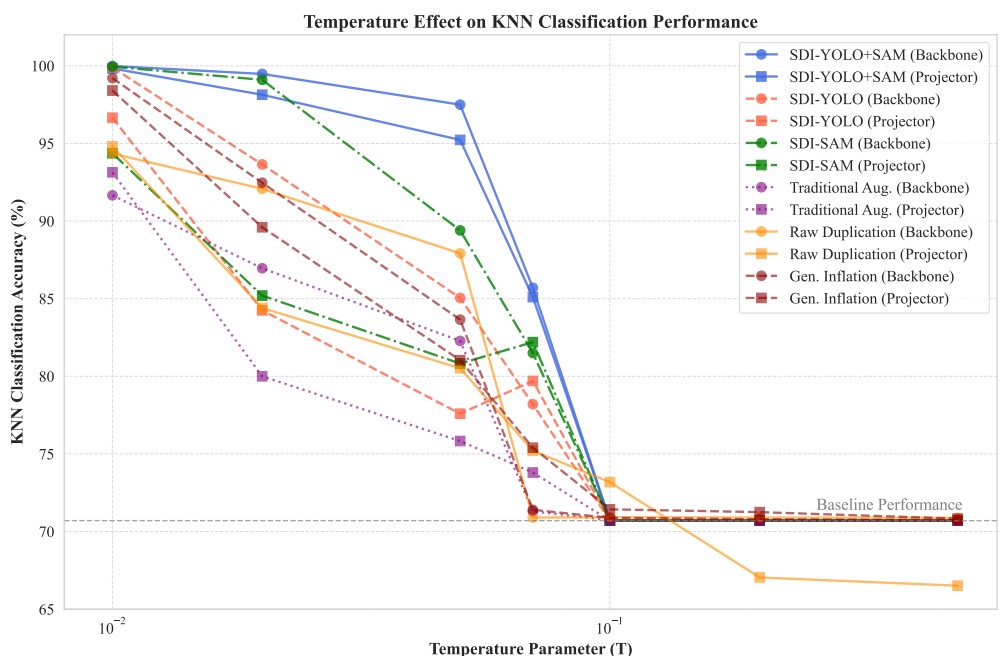

Figure 9: Effect of temperature on KNN accuracy (log scale $T$). SDI variants show higher accuracy at low temperature settings, with baseline methods declining more rapidly as $T$ decreases.

Figure 9 further illustrates the impact of temperature $T$ on KNN accuracy. The trend shows that SDI maintains consistently high performance at low temperatures compared to alternative augmentation methods.

## D.2 REPRESENTATION QUALITY ANALYSIS

Table 17 summarizes several structural and metric properties for learned representations. SDI achieves the highest intrinsic dimensionality and more favorable feature geometry according to uniformity and alignment metrics. Geometric and mixed invariance scores are also improved under SDI. Color invariance is slightly higher for standard augmentation, but differences are minor.

Table 17: Analysis of learned representations. **Bold** denotes best per metric.

| Metric | Standard Aug | **SDI** | Gen. Inflation |
|---|---|---|---|
| *Dimensionality (SimCLR)* | | | |
| Intrinsic Dim (TwoNN) | 35.2 | **48.7** | 39.3 |
| Intrinsic Dim (MLE) | 33.8 | **45.2** | 38.1 |
| *Feature Quality* | | | |
| Uniformity ($\downarrow$) | $-2.13$ | $-2.35$ | $-2.22$ |
| Alignment ($\downarrow$) | 0.37 | **0.28** | 0.33 |
| *Invariance Scores ($\uparrow$)* | | | |
| Geometric Transforms | 0.82 | **0.87** | 0.84 |
| Color Transforms | **0.89** | 0.86 | 0.88 |
| Mixed Transforms | 0.71 | **0.80** | 0.75 |
| *Topological Structure* | | | |
| Betti Curve AUC | 0.56 | **0.67** | 0.61 |

In addition, improvements in Betti curve AUC suggest that SDI-learned representations exhibit richer topological features.

Table 18 presents transfer performance for different scenarios. SDI shows higher accuracy in both cross-architecture and few-shot settings.

Table 18: Cross-architecture / few-shot transfer accuracy (%). **Bold** indicates best.

| Test Scenario | Standard Aug | **SDI** | Gen. Inflation |
|---|---|---|---|
| *Cross-architecture Transfer* | | | |
| ResNet-50 → ResNet-18 | 83.2 | **88.5** | 85.4 |
| ResNet-50 → MobileNetV2 | 82.3 | **87.1** | 84.6 |
| *Few-shot Classification* | | | |
| 1-shot | 45.3 | **53.8** | 49.2 |
| 5-shot | 68.7 | **74.5** | 71.4 |
| 20-shot | 82.6 | **87.3** | 84.5 |

Overall, SDI achieves the strongest results across different tasks and evaluations. The supplementary data above provide detailed quantitative evidence on feature robustness, representation geometry, and downstream transfer effectiveness, corroborating the primary conclusions in the main text.

## E  VISUALIZATION ANALYSIS

In this section, we present essential visualizations for three different datasets: CIFAR, TinyImageNet, and ImageNette. For each dataset, we showcase representative data augmentation samples. These visualizations help to understand the impact of augmentation methods on diverse image distributions.

### E.1  DATA AUGMENTATION SAMPLES

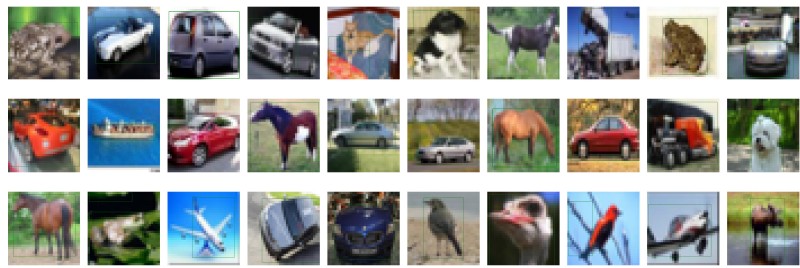

Figure 10: Examples of CIFAR images before and after different augmentation methods.

Figure 10 shows representative CIFAR images before and after applying various data augmentation techniques such as random cropping, flipping, and color jittering.

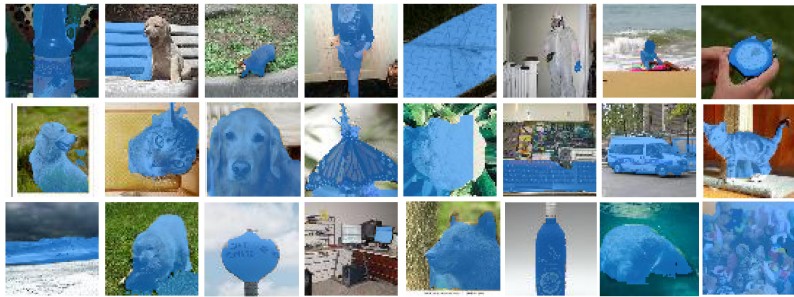

Figure 11: Examples of TinyImageNet images before and after different augmentation methods.

Figure 11 displays TinyImageNet samples processed with common augmentation operations. The visual comparison highlights how these methods alter the image properties and enhance data diversity.

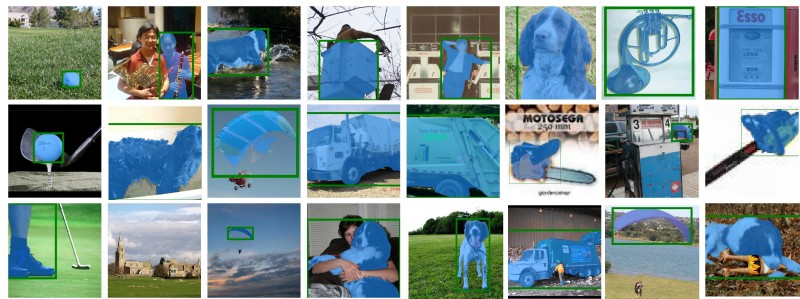

Figure 12: Examples of ImageNette images before and after different augmentation methods.

Figure 12 presents ImageNette images with and without augmentation. This visualization illustrates the visual changes introduced by the adopted data augmentation pipeline.

## E.2 ATTENTION HEATMAP VISUALIZATION

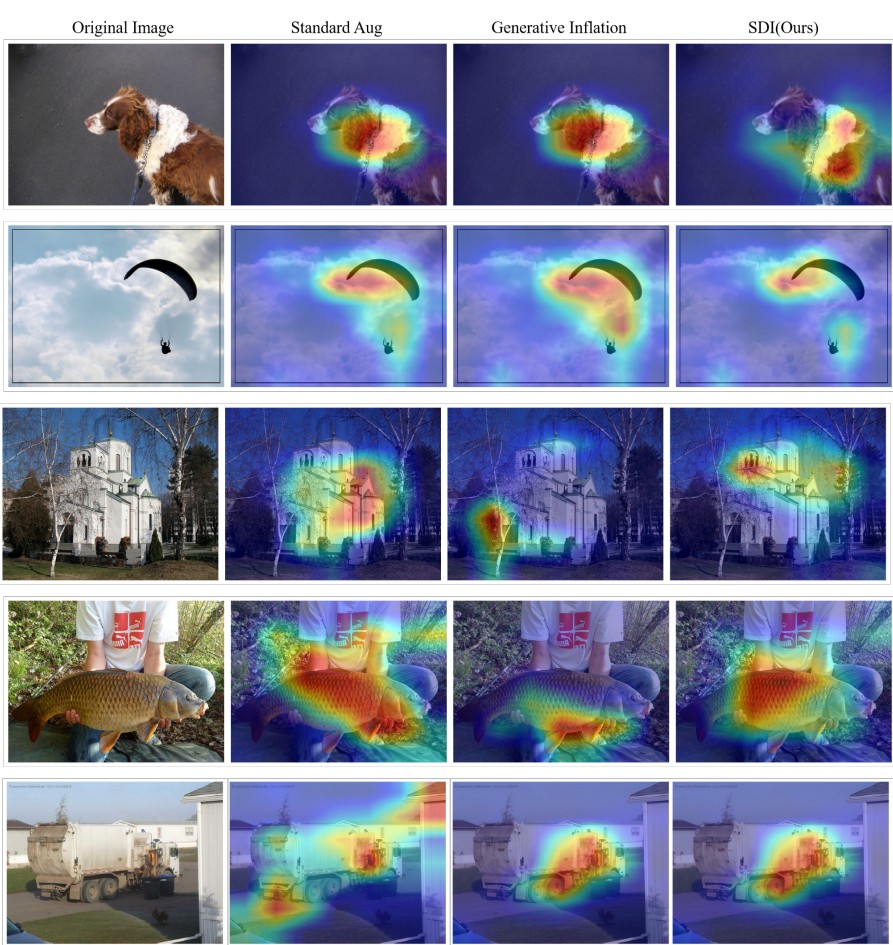

Figure 13: Grad-CAM attention heatmaps for selected samples.

Figure 13 displays Grad-CAM attention heatmaps generated for selected examples.

### E.3 FAILURE CASE ANALYSIS AND ROBUSTNESS

To address concerns regarding the reliance on external semantic oracles, we analyze the behavior of SDI in scenarios where the oracle provides incorrect or missing guidance. Our analysis demonstrates that the system exhibits graceful degradation and robust performance. As visualized in Figure 14, we identify two primary modes:

1. **Oracle Miss (Fallback Mechanism):** For highly abstract or low-quality images where the detector fails to find an object, SDI is designed with a protective fallback mechanism. As illustrated in the top row of Figure 14, the system reverts to applying standard random augmentation. This ensures the training pipeline remains stable, with performance gracefully degrading to the baseline level in such instances.

2. **Oracle Error (Structured Noise):** In rare cases, the oracle may "hallucinate" and place a bounding box on a background region. SDI directly uses this erroneous box as the augmented view. While this view lacks the primary object, it is not devoid of useful information. As can be observed (Figure 14, bottom row), such boxes often encapsulate coherent local textures or structured patterns (e.g., a patch of grass, a section of a wall). This provides a more structured, albeit semantically incorrect, learning signal compared to a completely random crop that might contain unstructured noise or disjointed object parts. The model is thus encouraged to learn representations of these coherent background features. Our strong overall empirical results confirm that the significant gains from correct guidance on the majority of the data far outweigh any potential misdirection from these infrequent, yet still structured, erroneous views.

This analysis confirms that SDI is a robust enhancement. It provides substantial benefits when the oracle is correct and degrades safely when it is not, validating its practical utility.

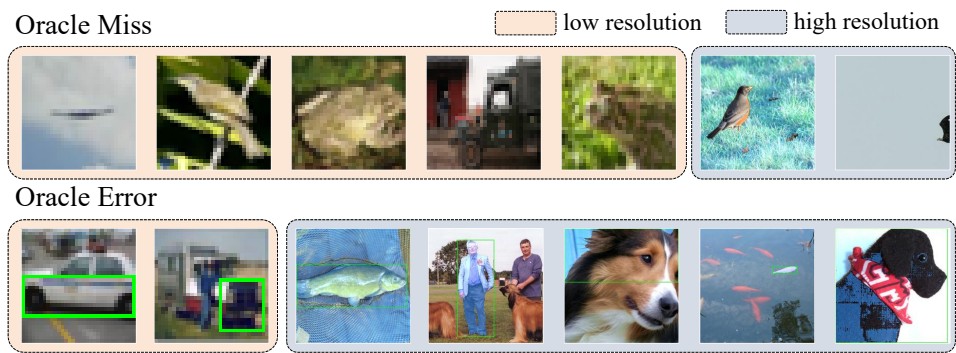

Figure 14: Visualization of SDI behavior in failure scenarios. **Top:** When the oracle fails to detect an object, SDI's fallback mechanism applies standard augmentation. **Bottom:** If the oracle hallucinates a box on the background, SDI uses this region directly. This view, while lacking the primary object, often contains structured information (e.g., textures), providing a different, useful, learning signal.

