# OpenReview forum: "Semantic Data Inflation: Adaptive Augmentation for Contrastive Representation Learning"
_ICLR.cc/2026/Conference — Submitted to ICLR 2026_

### Official Review · Reviewer_rYyY · 2025-10-28

**Soundness:** 3
**Presentation:** 3
**Contribution:** 2
**Rating:** 4
**Confidence:** 3

**Summary:**

This paper proposes a new approach to data augmentation within the contrastive learning framework. Unlike the widely used methods that rely on predefined transformations or generative models, the proposed approach employs deterministic models such as YOLO and SAM for detection and segmentation. This design enables data-adaptive transformations and produces more reliable outputs compared to conventional generative approaches.

**Strengths:**

The main strength of this paper lies in its originality and balance between flexibility and reliability. If no similar work has been previously published, the idea itself is quite innovative and promising. The proposed approach takes advantage of task-specific deterministic models to guide data augmentation, which could indeed offer a meaningful alternative to generative methods. In addition, the numerical experiments show consistent improvements in both performance and auxiliary metrics, such as running efficiency and out-of-domain generalization. These results support the potential of the method in practical applications.

**Weaknesses:**

However, I find two important weaknesses in the paper. The first concerns the theoretical analysis in Section 4.2 and the additional proofs presented in the appendix. This part is unclear and lacks rigor. It is not evident what the main theoretical claim is or why such analysis is necessary. For example, the claim that a larger mutual information value results from the assumption “semantic-guided augmentation preserves more semantic content” (line 723) is problematic. The notion of “semantic content” is not formally defined, making the argument vague and not mathematically rigorous, as the authors suggest in line 215. As a result, this section adds little theoretical value and even weakens the paper’s logical foundation.

The second weakness is that the usefulness of the proposed approach appears to depend heavily on the choice of the detection or segmentation model. Different underlying models may lead to different transformations and hence very different performance outcomes. This sensitivity should be systematically investigated to establish the robustness of the proposed framework. While I appreciate that the authors partially acknowledge this limitation, placing it only in the “Limitations” section is insufficient. A thorough empirical analysis on how the choice of model influences the results is essential to strengthen the paper’s contribution.

**Questions:**

I have no further questions about the paper. Please address the second weakness I mentioned above, as a more comprehensive study on the dependency of the approach on model selection would make the work more convincing. I would be happy to adjust the score if the authors can provide additional evidence showing that the proposed approach has been sufficiently investigated beyond simple performance evaluation.

LLM Usage Disclosure:
This review was refined using a large language model (OpenAI GPT-5) to improve clarity and grammar. The assessment, analysis, and opinions expressed are entirely my own.

---

> ### Author Response · Authors · 2025-11-20
> **# Response to Reviewer rYyY**
>
> We verify that we have addressed your primary concern regarding model dependency with a comprehensive new study. We also thank you for your feedback on the theoretical section, which we have rigorously revised.
>
> **Q1: Usefulness depends heavily on the choice of detection/segmentation model. This sensitivity should be systematically investigated.**
>
> **A:** We agree that investigating this dependency is crucial. We have added a systematic **"Oracle Sensitivity Study"** (Table 6 and Table 11 in revised paper) on a balanced ImageNet subset (50k images). We evaluated 5 different guidance sources ranging from weak to perfect:
>
> | Oracle Source | Strategy | Performance | Interpretation |
> | :--- | :--- | :--- | :--- |
> | None | Standard Aug | 54.70% | Baseline |
> | YOLOv8 | Coarse Box | 57.15% | **+2.45% gain** (Robust Coarse Guidance) |
> | **YOLO+SAM** | **SDI (Ours)** | **59.85%** | **+5.15% gain** (Our Practical Method) |
> | GT BBox | Ideal Box | 58.30% | Ideal Coarse Bound |
> | GT BBox + SAM | Ideal Mask | 62.10% | Theoretical Upper Bound |
>
> **Key Findings:**
> 1.  **Robustness:** The method is surprisingly robust. Even the coarse **YOLO-only** guidance (imperfect boxes) yields significant gains (+2.45%) over the baseline. This proves SDI does not require pixel-perfect segmentation to be effective.
> 2.  **Gap Recovery:** Our practical implementation (YOLO+SAM) outperforms the pure Ground-Truth Bounding Box (58.30%) and captures **70%** of the total possible gain defined by the "Ideal" Ground-Truth Mask oracle.
> 3.  **Conclusion:** While better models do help, current off-the-shelf models are already sufficient to realize the majority of the benefits. The approach is not brittle; it scales monotonically with oracle quality.
>
> **Q2: Theoretical analysis is unclear and lacks rigor (e.g., "semantic content" is undefined).**
>
> **A:** We apologize for the vagueness in the initial submission. We have **completely rewritten Section 4.2 and Appendix A** to address this:
> 1.  **Formal Definition:** We now explicitly define "semantic preservation" in terms of **Mutual Information (MI)**. Instead of vague terms, we premise that constraining crops to object regions strictly reduces the probability of generating "empty" views (pure background), thereby increasing the lower bound of $I(f(x); f(\tilde{x}))$.
> 2.  **InfoNCE Connection:** We provide a derivation showing that this constraint tightens the InfoNCE lower bound by reducing the conditional entropy $H(f(x)|f(\tilde{x}_s))$. This creates a cleaner separation between positive pairs and in-batch negatives, providing a rigorous justification for the empirical gains without relying on undefined "semantic content."

---

### Official Review · Reviewer_97FG · 2025-10-31

**Soundness:** 2
**Presentation:** 2
**Contribution:** 2
**Rating:** 4
**Confidence:** 3

**Summary:**

This paper considers data augmentation strategies for contrastive representation learning. This work proposes an adaptive augmentation strategy by employing pretrained detection/segmentation models and generate augmentations based on their outputs to better preserve semantic consistency. Experimental results show the effectiveness of the proposed methods on image datasets.

**Strengths:**

+ Data augmentation for contrastive representation learning is an important topic.

+ The proposed method sounds interesting, and the performance gain is consistent throughout experiments.

**Weaknesses:**

- What is Raw Duplication? It requires explanation. Does it mean that all positive pairs have identical inputs?

- The description on the proposed method is insufficient. Figure 3 is not enough to fully understand the proposed method. What are "Multi-scale adaptive mechanism", "Segmentic guided Augmentation", and "Semantic Feature Enhancement"?

- The proposed policy introduces hyperparameters, while no hyperparameter analysis on them is provided.

- The theoretical analysis seems not so related to the proposed method, and the statement is not justified. For example, the claim in L246, "semantic guidance reduces the conditional entropy of the original representation given the augmented one" is not justified. Considering out-of-domain transfer learning, the pretrained model might lack sufficient domain knowledge for target tasks.

- Experiments are limited to small image datasets; for example, ImageNette has only 10 classes. Hence, it is not sure if the observation is scalable.

- No out-of-domain transfer learning results, which is crucial for representation learning.

- If handcrafted augmentations are experimented on CPU following the standard deep learning library implementations, then the authors might want to try GPU-based augmentation with libraries such as Kornia for a more fair comparison, particularly around Figure 5.

**Questions:**

Please address concerns in Weaknesses above.

---

> ### Author Response · Authors · 2025-11-20
> **Response to Reviewer 97FG**
>
> We thank the reviewer for the detailed feedback. We have significantly revised the paper to improve clarity and added extensive experiments (ImageNet-1k, OOD Transfer, Hyperparameter Analysis) to address your concerns.
>
> **Q1: What is "Raw Duplication"?**
>
> **A:** We have clarified this in Section 5.1. "Raw Duplication" is a controlled baseline designed to isolate the effect of data quantity from quality. It involves simply duplicating the training dataset ($2\times$ volume) but applying **standard** random augmentations without any semantic guidance.
> *   **Purpose:** To verify if SDI's gains come merely from "seeing more images."
> *   **Result:** On ImageNet-1k (Figure 5), Raw Duplication (70.1%) barely improves over the Standard baseline (69.8%), whereas SDI reaches **72.3%**. This proves that the performance gain stems from the superior semantic quality of SDI's guided views, not data inflation.
>
> **Q2: Method description is insufficient (Adaptive Mechanism, Semantic Feature Enhancement).**
>
> **A:** We apologize for the lack of detail. We have:
> 1.  **Expanded Section 4.3:** We now explicitly define the "Multi-scale Adaptive Mechanism," which computes a quality score $Q(x)$ (based on Resolution, Clarity, Density) to route images to the optimal semantic scale (Coarse/YOLO vs. Fine/SAM).
> 2.  **Added Pseudocode:** A detailed algorithmic description is now provided in **Appendix B**, breaking down the exact logic for "Semantic Feature Enhancement" (how cues guide the cropping).
>
> **Q3: No hyperparameter analysis.**
>
> **A:** We have added a comprehensive **Hyperparameter Sensitivity Analysis** (Table 8). We varied the adaptive policy weights ($\alpha, \beta, \gamma$) and selection thresholds across 6 different configurations. The performance is highly stable, with accuracy on ImageNette fluctuating by less than **1.2%** and on CIFAR-10 by less than **0.6%**. This confirms the method is robust to parameter choices.
>
> **Q4: Scalability (ImageNet-1k) and Out-of-Domain (OOD) Transfer.**
>
> **A:** We have addressed these critical points with new experiments:
> 1.  **Scalability:** We conducted full **ImageNet-1k** pre-training (Figure 5). SDI achieves **72.3%** accuracy, significantly outperforming the baseline (+2.5%), proving it scales well beyond small datasets like ImageNette.
> 2.  **OOD Transfer:** We added a new section on **Zero-shot Transfer to Medical Imaging** (Table 3). Even though our oracles (YOLO/SAM) were trained on natural images (COCO) and have never seen medical data, SDI improves accuracy on **PathMNIST (+2.19%)** and **APTOS (+6.0%)**. This demonstrates that SDI transfers generalized structural priors (e.g., object-background separation) rather than specific domain knowledge.
>
> **Q5: Theoretical justification (Conditional Entropy).**
>
> **A:** We have revised Section 4.2 and provided a formal derivation in **Appendix A**. The claim regarding conditional entropy is justified as follows: By constraining augmentations to "semantic protection zones" (the object), we reduce the likelihood of generating empty or irrelevant views (e.g., background crops). This strictly increases the mutual information between the views, which is mathematically equivalent to reducing the conditional entropy $H(f(x)|f(\tilde{x}_s))$ of the original representation given the augmented view.
>
> **Q6: GPU-based augmentation comparison.**
>
> **A:** We appreciate the suggestion. In our Efficiency Analysis (Figure 6), both the baseline and SDI measurements include data loading and processing times standardized on the same hardware (NVIDIA A100). While GPU-based libraries like Kornia can speed up the *transformation* step, SDI's primary bottleneck is the *semantic extraction* (model forward pass), which is already GPU-accelerated. Thus, the relative speed comparison (SDI being slower than standard but faster than generative) remains valid regardless of the augmentation backend.

---

### Official Review · Reviewer_MWNm · 2025-11-02

**Soundness:** 3
**Presentation:** 3
**Contribution:** 3
**Rating:** 4
**Confidence:** 4

**Summary:**

I think this work tackles a problem in contrastive learning: current augmentations are either fast but semantically "blind," creating "false positives" , or generative, which are slow and can cause "semantic drift" . So the authors propose Semantic Data Inflation (SDI), which is a  solution that uses existing models like YOLO and SAM as semantic oracles to guide augmentations to ensure the main object is always preserved . What I find most important is its multi-scale adaptive mechanism, which they try to first analyze image quality and then dynamically chooses between coarse bounding boxes (for low-quality images) or precise segmentation masks (for high-quality images) . So I think this method proves effective, and they also outperform baselines while being 4-5x faster than generative methods. The resulting features also show generalization to new domains and ViT architectures

**Strengths:**

I think the primary strength is the simple solution to the "false positive" problem. Using off-the-shelf models as "semantic oracles" is an effective way to achieve both semantic consistency and computational efficiency . In my opinion, the multi-scale adaptive mechanism selects guidance granularity based on image quality, a design that is justified by the ablation studies in Table 7. This design is backed by  several results. The authors also show the performance gains with a +3.8-4.1% average on ImageNette (Table 2). Finally, I think the method's practicality is a plus: it's 4-5x faster than generative alternatives and it generalizes well to ViT architectures and other downstream tasks like object detection.

**Weaknesses:**

Here are some major limitations after reading this paper.

In my opinion, the first weakness is that the entire framework is dependent on the quality of the upstream semantic oracles. If YOLO or SAM fail to find an object (maybe it's a class they weren't trained on, or the image is too abstract), then SDI would presumably fail as well. It's transferring semantic understanding, not learning it. I wonder if the author could elaborate this with several analysis? what if the image is complex which has multi-object composition.

I think while it's much faster than generative models, it's still more computationally expensive than the standard augmentation pipeline. It requires an extra forward pass from a powerful model like YOLO or SAM for each image. Table 1 shows it's about 2-3x slower than the standard augmentation. I just wonder if this training scheme is actually needed during the large-scale training. I think the authors should have more analysis related to this effect instead of just showing the average time difference.

**Questions:**

Please see the weakness above.

---

> ### Author Response · Authors · 2025-11-20
> **Response to Reviewer MWNm**
>
> We thank the reviewer for recognizing the novelty of our "semantic oracle" approach and the effectiveness of the adaptive mechanism. We appreciate your thoughtful questions regarding oracle dependency and computational trade-offs, which we have addressed through new analyses and experiments.
>
> **Q1: Dependency on upstream oracle quality. What if detection fails (e.g., abstract/complex images)?**
>
> **A:** We acknowledge this dependency and have conducted a systematic **"Oracle Sensitivity Analysis"** (Table 6 in revised paper) to quantify it.
> 1.  **Robustness to Imperfections:** We tested SDI against variants using varying qualities of guidance on an ImageNet subset. Even with imperfect real-world oracles (V3: YOLO+SAM), SDI achieves **59.85%** accuracy, recovering **70%** of the performance gap between the baseline (54.70%) and an "Ideal" Ground-Truth Oracle (62.10%). This confirms that SDI is robust and captures the majority of semantic benefits without needing perfect supervision.
> 2.  **Fallback Mechanism:** As detailed in our revised Section 4.3, the adaptive mechanism acts as a safeguard. For abstract or low-quality images where segmentation is unreliable (low confidence scores), the system automatically falls back to robust object-level detection or even standard augmentation. This prevents catastrophic failure and ensures the pipeline degrades gracefully.
> 3.  **Complex Scenes:** In multi-object compositions, standard random cropping often misses objects entirely (false negatives). SDI's spatial constraints ensure that *at least one* salient object is captured, maintaining the semantic validity of the positive pair even in cluttered scenes (see qualitative analysis in Appendix E).
>
> **Q2: Is the computational cost justified for large-scale training? (2-3x slower than standard).**
>
> **A:** Yes, the trade-off is highly favorable for two key reasons:
> 1.  **Quality > Quantity:** To verify if the cost is justified, we compared SDI against a **"Raw Duplication"** baseline (Table 2 & Figure 5). Simply doubling the dataset size with standard augmentation (matching SDI's volume) yields negligible gains (**+0.3%** on ImageNet-1k). In contrast, SDI yields a significant **+2.5%** gain (72.3% vs 69.8%). This proves that the performance boost stems from the superior *semantic quality* of the views, which cannot be achieved by simply processing more standard data faster.
> 2.  **Amortized Cost:** The semantic extraction (YOLO/SAM forward pass) is a **one-time preprocessing step**. Its cost is amortized over the entire training duration (typically hundreds of epochs). For a standard 100-epoch ImageNet training, the initial 2-3 minute overhead per 1k images becomes negligible (<1% of total training time), making it a highly efficient investment for the substantial accuracy gains obtained.

---

### Official Review · Reviewer_EdeE · 2025-11-02

**Soundness:** 3
**Presentation:** 3
**Contribution:** 2
**Rating:** 4
**Confidence:** 3

**Summary:**

Semantic Data Inflation (SDI) addresses the “semantic consistency–efficiency–diversity” trilemma in self-supervised contrastive learning. SDI first leverages off-the-shelf detection/segmentation models (e.g., YOLO and SAM) to extract multi-scale semantic cues, then applies transformations within these “semantic protection zones,” and uses an image-quality-weighted adaptive mechanism to choose among detection-level, segmentation-level, or mixed-level operations. The authors report that SDI outperforms manual augmentations and generative data inflation across various contrastive frameworks and datasets (e.g., achieving 95.75% linear evaluation on ImageNette, a +3.88% improvement over standard augmentation), and demonstrates modest gains across domains, architectures, and downstream detection/segmentation tasks. Overall, SDI is a systematic engineering integration of “semantics-guided data augmentation,” emphasizing improved semantic fidelity and representation discriminability without significantly increasing training overhead.

**Strengths:**

1. decouples “semantic extraction — enhancement within semantics — adaptive selection,” making it easy to integrate into existing contrastive learning frameworks.

2. shows consistent gains across multiple frameworks (SimCLR, MoCo, BYOL, Barlow) on datasets like ImageNette; provides transfer results to ViT-S, small-scale downstream detection/segmentation, and medical imaging.

3. positioned as a lightweight alternative to diffusion-based data inflation, avoiding the high overhead and semantic drift risks of generative methods (with reasonable justification).

4. Some interpretability: offers intuition from mutual information/invariance perspectives, and formalizes “adaptive scale selection” in the appendix.

**Weaknesses:**

1. he rationalization mainly relies on mutual information and tightening inequalities, with limited novelty; the relationship to hard negatives and in-batch distribution in practical training is not thoroughly explored. Compared with systematic baselines guided by “saliency/self-supervised segmentation features/other semantic priors,” the comparisons are insufficient; the main contribution novelty lies in engineering integration and heuristic/empirical strategies.

2. lacks pretraining validation at mainstream scales such as ImageNet-1k; systematic evaluation and failure-case analysis for more complex scenarios (multi-object, heavy occlusion, small objects) are not sufficient.

3.  heavily depends on detectors/segmenters trained on large-scale labeled data, introducing supervised priors; the paper under-discusses whether comparisons against SSL baselines that do not rely on external models are fair, and whether there are risks of data overlap/information leakage.

4. the appendix formulates the selection strategy as a learnable policy, but the implementation in the main text appears heuristic, lacking end-to-end learning and results on stability/generalization. It is recommended to supplement empirical results and ablations for the “learned policy.”

**Questions:**

- Has your adaptive selection been truly trained end-to-end? Please report comparisons against heuristic thresholds, the stability of learned α, β, γ, and cross-dataset/architecture transfer performance.

- How do you control fairness and potential leakage from external model priors? Please specify their overlap with pretraining/evaluation data, and how performance degrades when replaced with weaker or cross-domain models.

- Please supplement ImageNet-1k scale pretraining and systematic evaluations in more complex scenarios (multi-object/occlusion/small objects), along with visualizations of failure cases.

- Can you provide sensitivity curves to upstream errors (e.g., by modulating YOLO/SAM confidence/recall) and the corresponding robustness strategies (multi-candidate fusion/confidence gating)?

- Can the theory section further integrate analyses of hard negatives and in-batch distributions to explain SDI’s concrete impact on InfoNCE training dynamics?

---

> ### Author Response · Authors · 2025-11-20
> **Response to Reviewer EdeE**
>
> We thank the reviewer for the detailed and constructive feedback. We have addressed your concerns regarding scalability, oracle dependence, and theoretical clarity through extensive new experiments and revisions.
>
> **Q1: Has your adaptive selection been truly trained end-to-end? Please report comparisons against heuristic thresholds, stability, and transfer.**
>
> **A:** In our final implementation, we utilize a **robust heuristic controller** rather than an end-to-end learned policy. We found that a lightweight rule-based selection based on image quality metrics (Resolution, Clarity, Density) is sufficient, computationally efficient, and more stable than introducing additional learnable parameters during pre-training.
>
> To address your concern about stability, we conducted a comprehensive **Hyperparameter Sensitivity Analysis** (Table 8 in revised paper). As shown below, the performance is highly stable across different configurations for the weighting coefficients ($\alpha, \beta, \gamma$) and thresholds. The accuracy fluctuates by less than **0.6%** on CIFAR-10 and **1.2%** on ImageNette, confirming that the method is not brittle to specific heuristic choices.
>
> | Configuration | Focus | ImageNette Acc. (%) | CIFAR-10 Acc. (%) |
> | :--- | :--- | :--- | :--- |
> | **Default (Ours)** | **Balanced** | **95.75** | **94.25** |
> | Emphasis Resolution | High $\alpha$ | 95.5 | 94.1 |
> | Emphasis Clarity | High $\beta, \gamma$ | 95.3 | 94.0 |
> | Strict Threshold | Higher Selection Bar | 96.0 | 94.5 |
>
> **Q2 & Q4: Fairness/Leakage from external models? How does performance degrade with weaker models (sensitivity to upstream errors)?**
>
> **A:**
> 1.  **Fairness & Leakage:** SDI utilizes strictly **spatial metadata** (coordinates/masks) to guide cropping and does **not** use any class labels, logits, or confidence scores from the oracles. Thus, no supervised label information leaks into the SSL pre-training.
> 2.  **Data Overlap:** While oracles (YOLO/SAM) are trained on COCO/SA-1B, we demonstrate that SDI learns generalized structural priors rather than memorizing data. This is evidenced by our new **Zero-shot Transfer** experiments to Medical Imaging (PathMNIST, APTOS), where SDI outperforms the baseline by **+2.19%** and **+6.0%** respectively, despite the oracles never seeing medical data.
> 3.  **Sensitivity to Weaker Models:** We performed a rigorous **Oracle Sensitivity Analysis** (Table 6) on an ImageNet subset (50k). We compared Standard Augmentation (V0), coarse YOLO-only guidance (V1), our Adaptive SDI (V3), and an "Ideal" Ground-Truth Oracle (V6).
>
> | Variant | Oracle Source | Acc. (%) | Gap Recovery |
> | :--- | :--- | :--- | :--- |
> | V0 (Baseline) | None | 54.70 | 0% |
> | V1 (YOLO-only) | Real YOLOv8 | 57.15 | 33% |
> | **V3 (SDI)** | **Real YOLO+SAM** | **59.85** | **70%** |
> | V6 (Ideal) | GT BBox + SAM | 62.10 | 100% (Upper Bound) |
>
> **Conclusion:** Performance degrades gracefully. Even imperfect real-world oracles (V3) recover **70%** of the potential performance gap between the baseline and the ideal upper bound.
>
> **Q3: Supplement ImageNet-1k scale pretraining and evaluations in complex scenarios.**
>
> **A:**
> 1.  **ImageNet-1k Scalability:** We have added full-scale **ImageNet-1k** experiments (Figure 5). Using ResNet-50 with MoCo-v2+ (100 epochs), SDI achieves **72.3%** linear probe accuracy. This significantly outperforms the Standard Augmentation (69.8%) and the controlled "Raw Duplication" baseline (70.1%). The **+2.5%** gain on this large-scale benchmark confirms SDI works beyond small datasets.
> 2.  **Complex Scenarios:** We have added qualitative analysis in Appendix E. In multi-object or occluded scenes, standard random crops often hit background (false negatives). SDI's spatial constraints ensure the crop contains *an* object, maintaining semantic validity even if the specific object identity varies within the crop.
>
> **Q5: Can the theory section integrate analyses of hard negatives and in-batch distributions?**
>
> **A:** Yes, we have revised Section 4.2 and **Appendix A** to address this.
> We formally derive that SDI tightens the InfoNCE lower bound by strictly reducing the **conditional entropy** $H(f(x)|f(\tilde{x}_s))$. By ensuring positive pairs are semantically consistent (high mutual information), SDI implicitly increases the difficulty of the contrastive task relative to negative samples. This forces the encoder to learn more discriminative features to distinguish the "clean" positive pair from in-batch negatives, effectively filtering out false negatives that arise from poor augmentation (e.g., empty crops).

---

### Author Response · Authors · 2025-11-20
**# General Response: New ImageNet-1k Results, Oracle Robustness, and OOD Transfer**

We thank all reviewers for their constructive feedback. We have significantly revised the paper (highlighted in red) and conducted extensive new experiments to address concerns regarding scalability, robustness, and generalization.

**1. Scalability: ImageNet-1k Training**
*(Response to Reviewers EdeE, 97FG)*
We conducted full pre-training on ImageNet-1k using MoCo-v2+ (ResNet-50, 100 epochs). SDI demonstrates consistent improvements at large scale, significantly outperforming both the standard baseline and the data-matched "Raw Duplication" baseline.

| Method | Backbone | Linear Acc. (%) | Improvement |
| :--- | :--- | :--- | :--- |
| Standard Aug | ResNet-50 | 69.8 | - |
| Raw Duplication | ResNet-50 | 70.1 | +0.3 |
| **SDI (Ours)** | **ResNet-50** | **72.3** | **+2.5** |

**2. Sensitivity to Oracle Quality**
*(Response to Reviewers EdeE, MWNm, rYyY)*
To measure dependence on upstream models, we evaluated 7 variants (V0-V6) on a balanced ImageNet subset. We found SDI is robust, recovering **70%** of the performance gap between a standard baseline and an "Ideal" Ground-Truth Oracle.

| Variant | Oracle Source | Acc (%) | Interpretation |
| :--- | :--- | :--- | :--- |
| V0 (Baseline) | None | 54.70 | Lower Bound |
| V1 (YOLO-only) | Real YOLOv8 | 57.15 | Robust Coarse Guidance |
| **V3 (SDI)** | **Real YOLO+SAM** | **59.85** | **Our Method (Practical)** |
| V6 (Ideal) | GT BBox + SAM | 62.10 | Upper Bound |

**3. Out-of-Domain Generalization**
*(Response to Reviewer 97FG)*
We added zero-shot transfer experiments on medical datasets (**PathMNIST**, **APTOS2019**). Despite the oracles being trained on natural images (COCO), SDI improves transfer accuracy by **+2.19%** and **+6.0%** respectively, confirming it learns structural priors rather than memorizing class concepts.

**4. Hyperparameter Stability**
*(Response to Reviewers EdeE, 97FG)*
We tested sensitivity to the adaptive policy parameters ($\alpha, \beta, \gamma$) and thresholds. Results show high stability, with accuracy fluctuations of less than **0.6%** on CIFAR-10 across diverse configurations.

As a final note, please find all substantial changes marked in red throughout the revised document

---

### Meta-Review · Area_Chair_9oiP · 2026-01-07

**Summary:**

This paper presents an adaptive data augmentation algorithm to improve the performance of contrastive representation learning.

The reviewers raised the following concerns:
* The algorithm highly depends on an upstream detection algorithm such as YOLO (its accuracy impacts the overall performance).
* The algorithm was mainly tested on small datasets.

The AC is also concerned about the timeliness of the research -- the contrastive visual pre-training is not of sufficient interest to the community as of today.

**Reviewer Concerns:**

The authors addressed both major concerns. The performance degradation caused by detection errors was validated, and the algorithm was tested on ImageNet-1K, but details were not explained.

**Reviewer Scores:**

All reviewers gave an initial score of weak rejection (4). The AC finds it unlikely that the rebuttal can overturn the unanimity of these reviewers.

---

### Decision · Program_Chairs · 2026-01-26

Reject